# Applicability of the Electrochemical Oxygen Sensor for In-Situ Evaluation of MgO Solubility in the MgF₂–LiF Molten Salt Electrolysis System

**Youngjae Kim** [1] , **Junsoo Yoo** [1] **and Jungshin Kang** [2,3,*]

[1] Mineral Resource Research Division, Korea Institute of Geoscience and Mineral Resources (KIGAM), Daejeon 34132, Korea; youngjae.kim@kigam.re.kr (Y.K.); yjs7620@kigam.re.kr (J.Y.)

[2] DMR Research Center, Korea Institute of Geoscience and Mineral Resources (KIGAM), Daejeon 34132, Korea

[3] Resource Recycling Department, University of Science & Technology, Daejeon 34113, Korea

\* Correspondence: jskang@kigam.re.kr; Tel./Fax: +82-42-868-3670

**Abstract:** The measurement and evaluation of MgO solubility in the molten fluoride system is of significant importance in the recently proposed magnesium electrolysis reduction process. In the present study, an in-situ quantitative method of evaluating the concentration of dissolved MgO in molten fluoride is proposed. The MgO solubility in the 32.8MgF₂–67.2LiF system was measured at 1083 and 1123 K using a combustion analyzer. MgO saturation was achieved in under 2 h, and higher solubilities were observed as the temperature increased. Thermodynamic assessment was carried out in order to ascertain the applicability of the electrochemical oxygen sensor, which indicated that the logarithm of oxygen concentration in molten fluoride has a linear relationship with the measured electromotive force (EMF) potential. The EMF potential of the controlled MgO concentration was measured, and a straight calibration line was obtained, describing the relationship between the measured EMF and the logarithm of MgO concentration. From the obtained calibration line, MgO concentration in the 0.4 wt% MgO was calculated. The calculated value was 0.44 wt% that was in excellent accordance with the controlled MgO concentration of 0.4 wt%, verifying the practical applicability of electrochemical oxygen for the in-situ monitoring and evaluation of MgO solubility in the electrolysis magnesium reduction process.

**Keywords:** magnesium electrolysis reduction; electrochemical oxygen sensor; MgO solubility; MgF₂–LiF; EMF cell

---

## 1. Introduction

Owing to its high strength-to-weight ratio, precision machinability, and applicability in various alloy systems, magnesium is widely used as a structural metal and alloying element [1–3]. Approximately 70% of the magnesium used is in metallurgical applications [2]. Magnesium is known to improve the mechanical properties, strength, and corrosion resistance of non-ferrous metals such as aluminum and zinc when alloyed with these materials [2–5]. The addition of a small amount of magnesium in the iron- and steel-making process can enhance the mechanical properties of the steel product, including the ductility, strength, and toughness [2]. An increase in the requirement for lightweight materials by the automobile industry as a means of obtaining higher fuel efficiency is expected to result in a higher demand for magnesium [6,7]. The annual growth rate in the demand for magnesium is expected to reach 4.5–8% by 2020 [8]. Magnesium has been identified as a critical raw material by the European Union as a result of its high economic impact and the risks associated with maintaining adequate supplies [9].

As a result of the increasing demand for magnesium, considerable attention has been paid to the process by which magnesium is extracted. Two processes are currently used in the commercial extraction of magnesium; electrolytic and thermochemical reduction [2]. The Pidgeon process is the main thermochemical process that is used for reduction. Approximately 85% of the current global production of magnesium is via the Pidgeon process in China [10]. However, the energy consumed by magnesium production using this method is relatively high [2,10] because of the requirement for high reaction temperatures (approximately 1473 K) and vacuum conditions (13 Pa) [2]. The current generation of magnesium using the Pidgeon process requires horizontal steel tubes known as retorts [10] and the magnesium is reduced in batches, which results in lower production efficiency. The process also generates large amounts of carbon dioxide and particulate matter (PM2.5), which is problematic from an environmental perspective [10,11]. The continuous extraction of magnesium using electrolytic reduction is more efficient in terms of energy and generates low $CO_2$ emissions, meaning that this method is significantly more economical than thermochemical reduction. However, toxic gas ($Cl_2$) is generated during the electrolytic process, which has led to environmental concerns [12].

In order to resolve the environmental problems that result from conventional processes, several novel magnesium extraction processes have been proposed. The solid oxide membrane (SOM) process is considered an environmentally friendly metal extraction technique with high energy efficiency and low capital costs [13]. Owing to its simplicity and energy efficiency, SOM is often utilized for the extraction of high-purity magnesium, silicon, titanium, tantalum, and aluminum [12–18]. In the SOM process, the liquid–metal anode is separated from a molten fluoride bath containing dissolved target metal oxide by the oxygen-ion-conducting material: yttria-stabilized zirconia (YSZ). The targeted metal ion is reduced at the cathode during electrolysis, while the oxygen ions migrate to inside the YSZ membrane and form oxygen gas at the liquid–metal anode. SOM does not generate chlorine or carbon dioxide gases and the cost of magnesium extraction is much lower than that using conventional electrolysis or the Pidgeon process [13]. However, the YSZ membrane degrades over time, leading to a significant decrease in the current efficiency, and the relatively high operating temperatures are regarded as a major hurdle for the practical application of SOM [13,19]. Kang et al. [19] recently proposed an advanced electrolysis reduction process in which chlorine gas is not generated during the reduction of magnesium. In this process, MgO that is dissolved in a molten fluoride bath is reduced, forming a magnesium alloy with the molten metal cathode. An inert electrode is used, meaning that there is no reaction between the oxygen and the anode material; therefore, pure oxygen gas forms at the surface of the anode. Magnesium of high purity can be obtained from magnesium alloys via this method.

Both SOM and the method developed by Kang are based on the electrolytic reduction of magnesium from MgO that has been dissolved in a molten fluoride bath. Evaluating the solubility of MgO in molten fluoride is therefore important for the practical utilization of these methods. Recently, several studies have been carried out regarding MgO solubility in various molten fluoride systems [20–22]. However, in spite of the importance of monitoring MgO dissolution [13], no in-situ evaluation technique has been established for use in the process of magnesium reduction via electrolysis. Although Guan et al. [13] proposed a method in which the dissolution of magnesium in molten salts can be assessed by measuring the electronic transference number of a flux between two electrodes, a quantitative evaluation technique for MgO dissolution in molten fluoride has not yet been reported. The electrochemical oxygen sensor has been practically adopted for the evaluation of oxygen solubility and its related thermodynamic potential in the steelmaking and glassmaking processes [23–25]. Therefore, in the present study, a quantitative assessment of MgO dissolution was carried out using an electrochemical oxygen sensor. The MgO solubility in the 32.8(mol%)MgF$_2$–67.2(mol%)LiF system was measured at 1123 K. The electromotive force (EMF) in the 32.8MgF$_2$–67.2LiF system was also measured with varied MgO concentrations in order to establish a calibration line between the logarithm of dissolved MgO concentration and EMF potential. Finally, the EMF of the controlled MgO concentration system

was measured and the MgO solubility was quantitatively determined by interpolation using the calibration line.

## 2. Experimental Procedures

### 2.1. Sample Preparation and MgO Solubility Analysis

The reagent grade $MgF_2$ and LiF were dried at 453 K for 72 h in the vacuum oven in order to remove the moisture. The sample was then weighed and mixed in an agate mortar in order to evaluate the solubility of the MgO in the molten fluoride. An amount of 3 g of the prepared mixture was melted in the MgO crucible (inner diameter: 13 mm, outer diameter: 18 mm, height: 50 mm) located within the $MoSi_2$-equipped vertical tube furnace. During the experiment, argon gas was supplied at 300 $cm^3$/min using a MFC controller and the temperature was precisely controlled with a calibrated B-type thermocouple and a PID controller. After the experiment, the sample was unloaded and quenched by blowing Ar gas. The obtained sample was then separated from the MgO crucible and the surface ground with a rotary tool in order to remove the attached MgO. The sample was then crushed and several pieces were collected from the central part of the sample for oxygen analysis.

Because the system itself contains $MgF_2$, the solubility of MgO cannot be measured by analyzing the concentration of magnesium ion. The solubility of MgO in the $MgF_2$-containing fluoride system has previously been measured by analyzing the concentration of oxygen, which originates mainly from the dissolved magnesium oxide [22]. Lee et al. [22] found that the concentration of oxygen in a dehydrated fluoride sample was less than 100 ppm, implying that oxygen concentration is directly related to the concentration of dissolved magnesium oxide. In this study, the oxygen concentration was determined using a combustion analyzer (TCH-600, LECO, St. Joseph, MI, United States) and calibrated using a standard sample containing 0.353 wt% oxygen prior to the analysis. Approximately 0.1 g of sample was weighed and placed in a high purity tin capsule (diameter: 5.0 mm, height: 13 mm), after which it was placed in the graphite crucible and heated in an impulse furnace. Using non-dispersive infrared cells, the oxygen concentration was analyzed by detecting the CO and $CO_2$ gases resulting from the reaction between the oxygen and the graphite crucible. The measured oxygen concentration (wt% O) was converted to MgO concentration (wt% MgO) using Equation (1) [22]:

$$(\text{wt\% MgO}) = \frac{M_{MgO}}{M_O}(\text{wt\% O}) \tag{1}$$

where $M_{MgO}$ and $M_O$ are the molecular weights of MgO and oxygen, respectively. The measurement was repeated more than three times, from which the average was calculated.

### 2.2. Electromotive Force (EMF) Measurement

For the EMF measurement, a 68 g sample was prepared by mixing 32.8$MgF_2$–67.2LiF with 0.2, 0.4, or 0.6 wt% MgO in an agate mortar. The mixture was placed in a Pt-10%Rh crucible (inner diameter: 40 mm, outer diameter: 41 mm, height: 65 mm) and loaded in the vertical tube furnace (detailed specifications concerning the furnace are given in the previous section). Measurement of the EMF potential of the MgO saturated 32.8$MgF_2$–67.2LiF system was carried out by placing a MgO cylinder (inner diameter: 13 mm, outer diameter: 18 mm, height: 20 mm) in the center of the Pt crucible containing the 32.8$MgF_2$–67.2LiF powder mixture. The sample was then heated at 1123 K under an argon atmosphere. These conditions were maintained for 6 h, after which the oxygen sensor and Pt wire were immersed in molten fluoride. The EMF was measured using a digital multimeter (34461A; Keysight, Santa Rosa, CA, United States) and recorded using Benchvue software. A schematic of the EMF measurement apparatus used is produced in Figure 1.

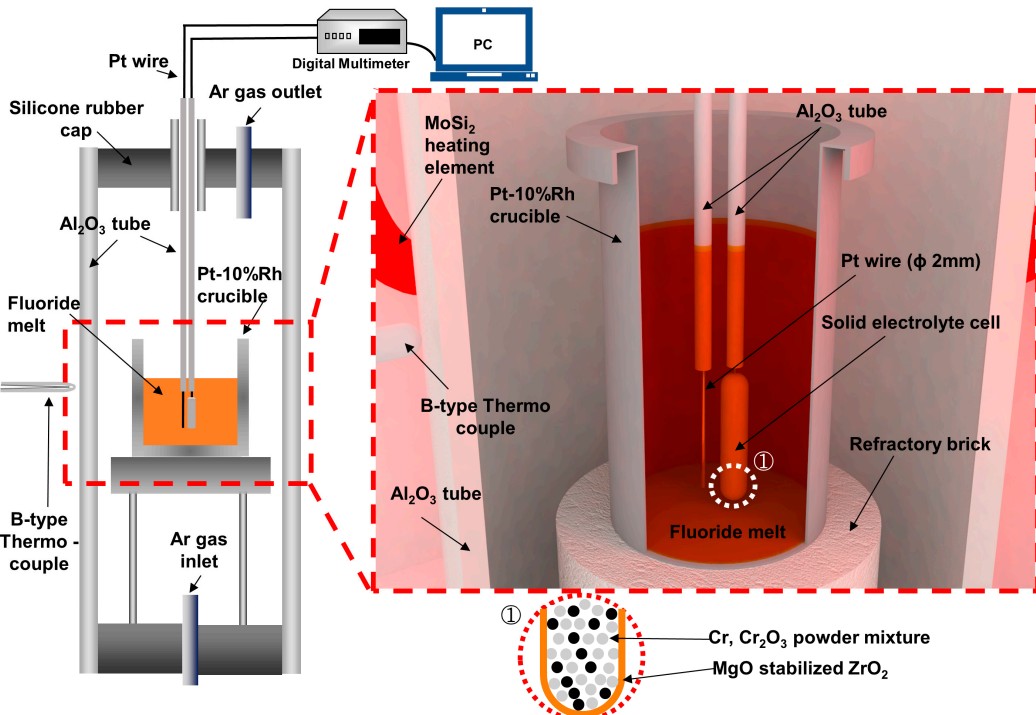

**Figure 1.** Schematic of the experimental apparatus used for measurement of the electromotive force (EMF) potential. The small inset in the dashed circle(①) is a cross-sectional image of a reference electrode cell.

*2.3. Thermal Property Analysis*

Thermogravimetry-differential thermal analysis (TG-DTA, DTG-60H, Shimadzu, Tokyo, Japan) was performed to determine the melting temperature of the system. A 20 mg mixture of $32.8MgF_2-67.2LiF$ powder was placed in a platinum pan. Aluminum oxide was used as a reference material during this experiment, with argon gas flowing at a controlled rate of 20 cm$^3$/min. The sample was initially heated to 473 K at a rate of 10 K/min, after which this temperature was maintained for 30 min in order to remove the moisture. The sample was then heated to 1123 K at a rate of 5 K/min. Heat flow was recorded as thermal voltage (μV) during the experiment.

## 3. Results and Discussion

*3.1. Determination of the Saturated MgO Concentration in Molten Fluoride*

During the process of electrolysis reduction, the MgO is reduced and magnesium metal is obtained in a liquid state. A low temperature is therefore required in order to maintain the low vapor pressure and prevent vaporization of the magnesium metal. The eutectic composition $MgF_2-LiF$ system was selected for use in the evaluation of MgO solubility. As seen in Figure 2, the liquidus temperature of the $32.8MgF_2-67.2LiF$ system was determined at approximately 1010 K, which is in excellent accordance with a previously reported eutectic temperature of 1008 K for the $MgF_2-LiF$ system [26].

In order to investigate the effect of time on MgO solubility, the MgO solubility of the $32.8MgF_2-67.2LiF$ system was measured while varying the experimental time. As shown in Figure 3, equilibrium can be achieved in the MgO and $32.8MgF_2-67.2LiF$ system in less than 2 h. In addition, equilibrium was achieved in the $46.5MgF_2-46.5CaF2-7LiF$ system [22] in less than 3 h at temperatures lower than 1273 K. Ito and Morita [27], who studied MgO solubility in the molten $MgCl_2$ and $CaCl_2$ systems, also reported an equilibrium time of less than 2 h. Considering that equilibrium between solid MgO and oxide systems takes more than 8 h [28], the equilibrium time of 2 h for molten fluoride obtained in this study is significantly short.

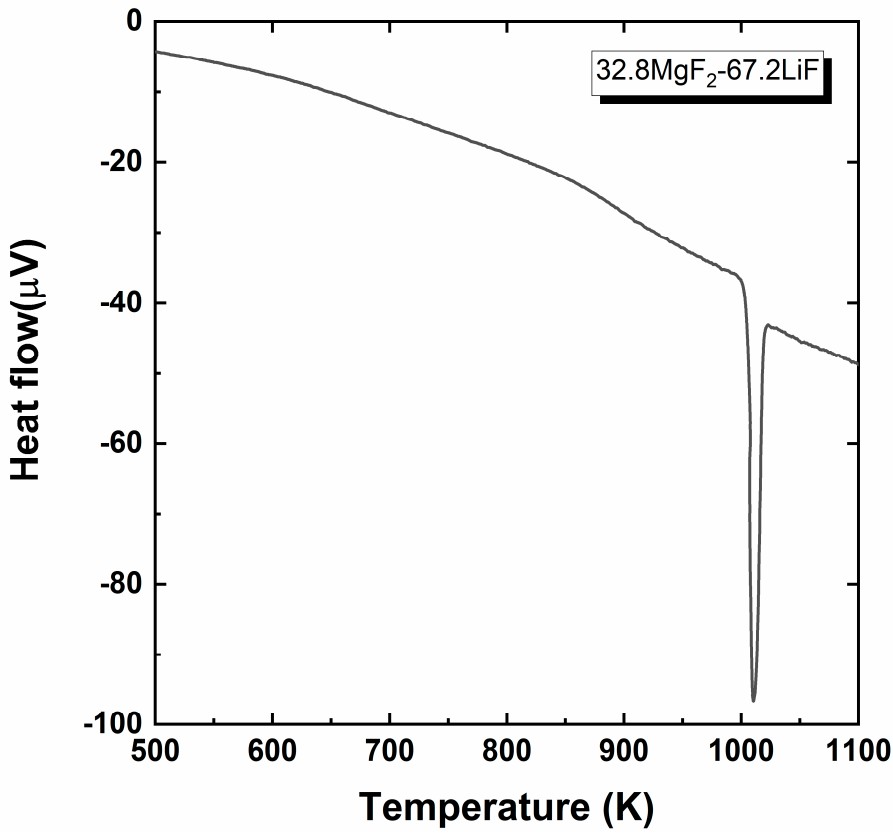

**Figure 2.** Differential thermal analysis (DTA) thermal curve of the 32.8MgF2–67.2LiF system.

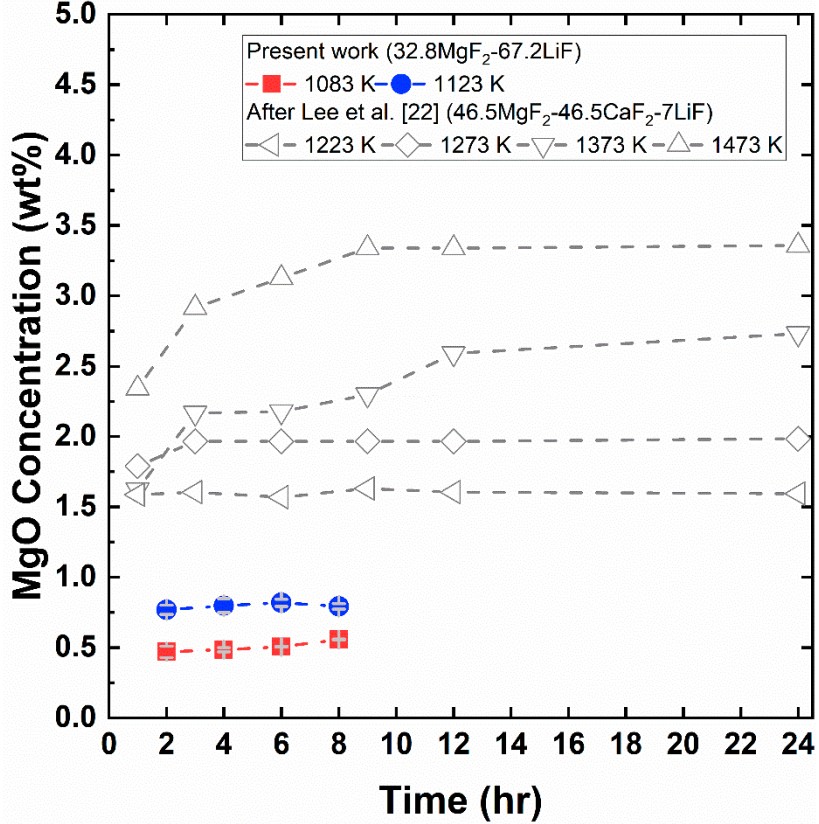

**Figure 3.** The solubility of MgO in the 32.8MgF2–67.2LiF and $MgF_2$–$CaF_2$–LiF [22] systems with varied reaction times.

The effects of temperature on MgO solubility in the molten fluoride system can be seen in Figure 4. The solubility of MgO in the 32.8MgF$_2$–67.2LiF system is determined at 0.56 and 0.79 wt% at 1083 and 1123 K, respectively. A similar dependence on temperature has been observed in other systems. As shown in Figure 4, MgO solubility increases in 46.5MgF$_2$–46.5CaF$_2$–7LiF and 45MgF$_2$–45CaF$_2$–10NaF systems as the temperature is increased [22]. In addition, an increase from 1.0 wt% to 1.9 wt% has also been observed in MgO solubility as a result of increasing the temperature of MgCl$_2$ from 1073 to 1373 K [27].

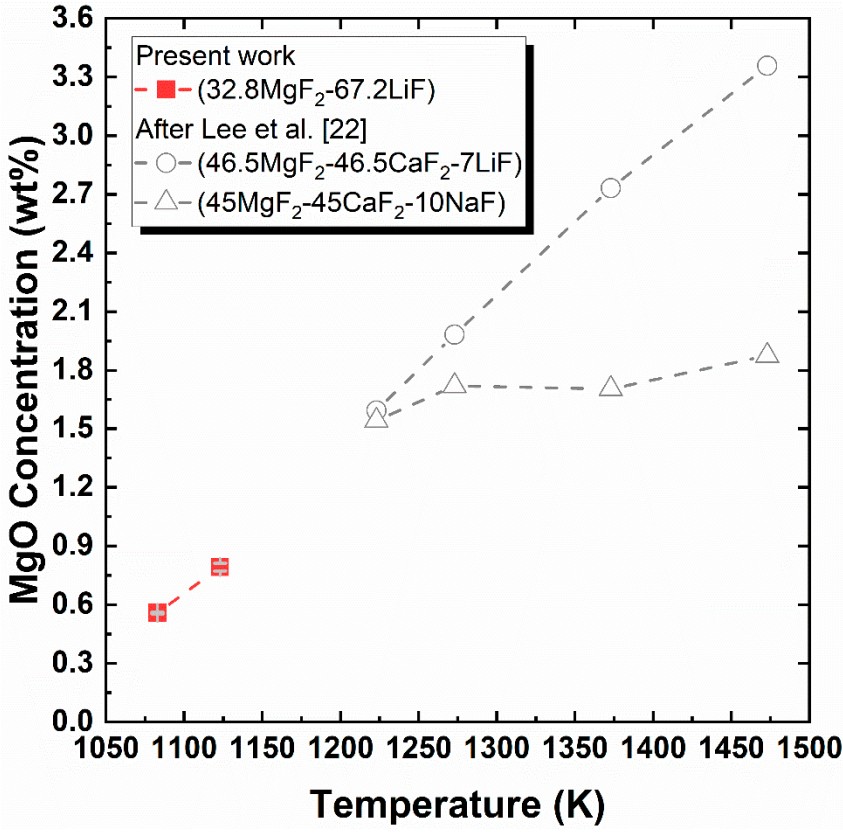

**Figure 4.** Relationship between temperature and MgO solubility in the 32.8MgF2–67.2LiF system and the MgF$_2$–CaF$_2$–(LiF or NaF) systems [22].

According to Cherginets [29], the solubility of oxide in the molten halide system is dominantly affected by interaction between the components that comprise the solvent. Molten halides characteristically form complexes via an association between the metal cations and the halide anions. Highly stable complexes with strong interactions result in lower oxide solubilities. This infers that the interaction between complexes decreases as the temperature increases, resulting in the higher oxide solubility observed in the molten halide system.

It should be noted that MgO solubility in the system used in this study is in accordance with the extrapolated MgO solubility of the 46.5MgF$_2$–46.5CaF$_2$–7LiF system [22] portrayed in Figure 4. It is apparent that neither the addition of CaF$_2$ or a decrease in the LiF concentration significantly affects MgO solubility, and the effect that temperature has on MgO solubility is more significant. This infers that the stability of halide complexes within the MgF$_2$–LiF system is not affected by compositional changes in LiF or the addition of CaF$_2$, but that the system is predominantly affected by changes in the temperature.

*3.2. Application of the Electrochemical Oxygen Sensor for the Determination of MgO Solubility in the Molten Fluoride System*

The potentiometric titration method has been widely adopted for the determination of oxygen solubility in various solutions [29,30]. This technique is based on establishing a linear relationship between the potentiometric cell EMF and the concentration of oxide ions. The oxygen concentration within a solution can be determined from this linear relationship, even at extremely low oxygen concentrations [29]. Due to the very simple process and the high reproducibility of EMF measurement, along with the possibility of in-situ measurement, the potentiometric titration method has a significant advantage over other oxygen solubility measurement techniques [29].

The EMF cell used in the present study can be expressed using the following Equation:

$$Pt/Cr + Cr_2O_3 \; // \; ZrO_2(+MgO) \; // \; Melt, \; [O]/Pt \tag{2}$$

From the left of this equation, the platinum reference electrode is kept at a constant oxygen partial pressure via the equilibrium between Cr and $Cr_2O_3$. The powdered mixture of Cr and $Cr_2O_3$ is separated from the melt by the magnesia stabilized zirconium oxide (MSZ; $ZrO_2(+MgO)$;) cell that has the unit oxide transport number [24]. Electrochemical oxygen transport occurs between the reference electrode and the melt in the EMF cell, meaning that the cell EMF ($E$) can be expressed using the following equation.

$$E = \frac{RT}{F} \ln \frac{P_{O_2(ref)}^{1/4} + P_{\theta}^{1/4}}{P_{O_2(salt)}^{1/4} + P_{\theta}^{1/4}} \tag{3}$$

where $R$ is the gas constant (8.3144 J/mol K), $T$ is temperature (K), $F$ is the Faraday constant (94,687 coulombs/mol), $P_{O_2(ref)}$ is the oxygen partial pressure of the reference electrode, $P_{O_2(salt)}$ is the partial pressure of oxygen in molten fluoride, and $P_{\theta}$ is a parameter that is related to the partial electronic conductivity of the solid ionic conductor [25]. For magnesia stabilized zirconia, $P_{\theta}$ can be expressed using the following equation [23]:

$$logP_{\theta} = 23.71 - 69,640/T \tag{4}$$

The oxygen partial pressure of the reference electrode ($P_{O_2(ref)}$) is determined from the reaction between Cr and $Cr_2O_3$, as described in Equation (5).

$$\frac{2}{3}Cr(s) + \frac{1}{2}O_2(g) = \frac{1}{3}Cr_2O_3(s) \tag{5}$$

From the equilibrium relation, the Gibbs free energy change of reaction (4) can be obtained using Equation (6) [23]:

$$\Delta G^0 = -RTln\frac{a_{Cr_2O_3}^{1/3}}{a_{Cr}^{2/3} P_{O_2(ref)}^{1/3}} = -370,000 + 82.3T \tag{6}$$

As the activity of Cr and $Cr_2O_3$ is unity in the reference cell, the oxygen partial pressure of the reference electrode ($P_{O_2(ref)}$) can be calculated at any given temperature using Equation (6).

By re-arranging Equation (3), the oxygen partial pressure of the fluoride melt can be described using the following Equation (7).

$$P_{O_2(salt)} = \left[ \frac{P_{O_2(ref)}^{1/4} + P_{\theta}^{1/4}}{\exp\left(\frac{EF}{RT}\right)} - P_{\theta}^{1/4} \right]^4 \tag{7}$$

Since the oxygen dissolved in the melt is never at equilibrium with the gaseous oxygen [23], the measured $P_{O_2(salt)}$ is the oxygen partial pressure equilibrated with the activity of dissolved oxygen, which can expressed via the following reaction.

$$\frac{1}{2}O_{2(salt)} = [O]_{salt} \tag{8}$$

In the equilibrium state, the Gibbs free energy change of reaction (8) can be expressed using Equation (9).

$$\Delta G^0 = -RTln\frac{a_{[O]\ salt}}{P^{1/2}_{O_2(salt)}} \tag{9}$$

$$a_{[O]\ salt} = f_{[O]\ salt} \cdot X_{[O]\ salt} \tag{10}$$

Since the activity of oxygen in the molten fluoride ($a_{[O]\ salt}$) is a function of the activity coefficient of oxygen in molten fluoride ($f_{[O]\ salt}$) and the mole fraction of oxygen in molten fluoride ($X_{[O]\ salt}$), Equation (9) can be re-written (11):

$$P^{1/2}_{O_2(salt)} = \frac{f_{[O]\ salt}}{\exp(-\Delta G^0/RT)} X_{[O]\ salt} \tag{11}$$

Considering the limited oxygen solubility in the molten fluoride system, it can be assumed that dilute solution of oxygen in a molten $MgF_2$–LiF system obeys Henry's law. Thus, the coefficient activity of oxygen in molten fluoride ($f_{[O]\ salt}$) can be considered unity. In addition, the change in the Gibbs free energy ($\Delta G^0$) is a function of temperature in $32.8MgF_2$–67.2LiF. Therefore, it can be deduced that the oxygen partial pressure measured by the EMF cell is linearly proportional to the concentration of dissolved oxygen in the molten fluoride at a fixed temperature.

$$P^{1/2}_{O_2(salt)} \propto X_{[O]\ salt} \tag{12}$$

As discussed in Equation (3), $P_{O_2(salt)}$ is calculated by the measurement of EMF potential. Therefore, the linear relationship can be expressed by the function of EMF and $\ln X_{[O]\ salt}$;

$$EMF \propto \ln X_{[O]\ salt} \tag{13}$$

The raw data for the EMF measurements are given in Figure 5, which indicates that the EMF initially increases dramatically as the cell is immersed in the molten fluoride bath. After approximately 15–20 s, the EMF became constant as the oxygen potential of the reference electrode ($Cr$-$Cr_2O_3$) attained equilibrium. After a few tens of seconds, the EMF gradually decreased as a result of the damage caused to the MSZ used in the electrode cell. The voltage at the first plateau was therefore selected as the oxygen potential measurement in this study, as described by the red line.

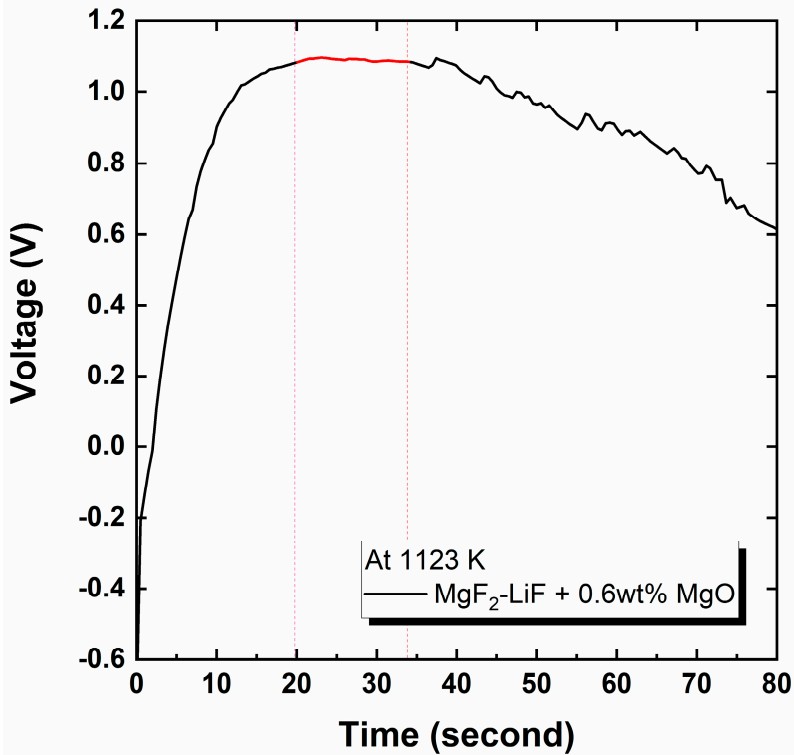

**Figure 5.** The raw data obtained during the EMF potential measurement. The red line indicates the potential chosen for the calculation of oxygen partial pressure.

The linear relationship between the measured EMF of the molten fluoride system with the logarithm of MgO concentration is shown in Figure 6. Based on the linear relationship explained in Equation (13), the calibration line was obtained by fitting with the least square method. The calibration line was established by using three systems: 0.2 wt%, 0.6 wt% of MgO, and MgO saturated systems. After EMF measurement in the MgO saturated system, MgO concentration was determined by using the combustion analyzer. The analyzed MgO concentration was 0.781 wt%. As discussed in the previous section, the measured MgO solubility in the 32.8MgF$_2$–67.2LiF system was 0.793 wt% at 1123 K, indicating that MgO saturation was achieved in the experiment involving EMF measurement.

As shown in Figure 6, the EMF potential of the 0.4 wt% MgO system was plotted on the present calibration line (red circle). From the calibration line, the calculated MgO concentration was 0.44 wt% that is in excellent accordance with the controlled MgO concentration of 0.4 wt%. Therefore, it can be concluded that the electrochemical oxygen sensor is practically feasible for the evaluation of MgO solubility in the MgF$_2$–LiF molten salt electrolysis system. By increasing the number of samples for establishing the calibration line, the accuracy for determining the MgO concentration in the MgF$_2$–LiF system would be enhanced.

Although the durability of MSZ used in the present solid electrolyte cell was not found to be suitable for long-term measurement, there are forms of MSZ and YSZ that are highly durable in fluoride melt at high temperatures [31]. By adopting these types of MSZ or YSZ in solid electrolyte cells, the measurement and monitoring of dissolved MgO concentration could be possible in-situ during the electrolytic magnesium reduction process.

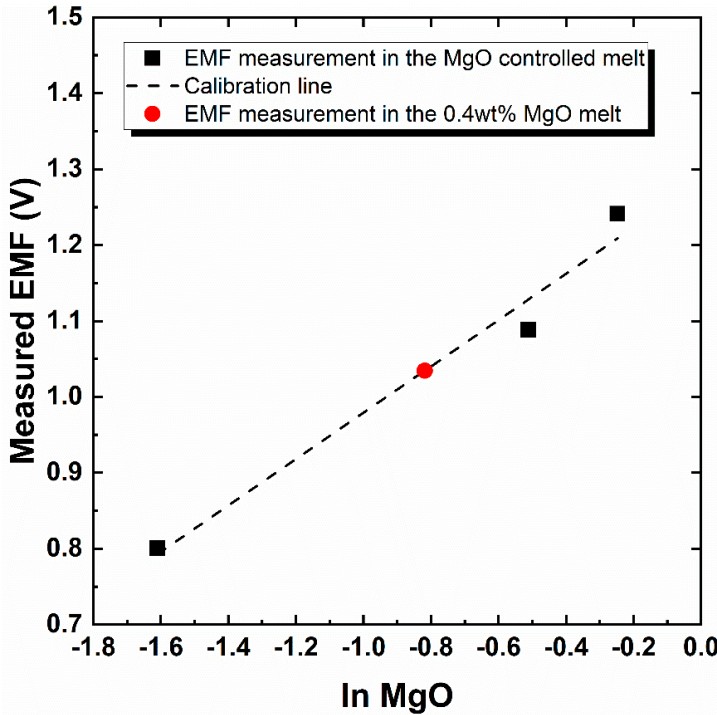

**Figure 6.** Relationship between the measured EMF and logarithm of MgO concentration. The black dashed line is the calibration line used for predicting MgO concentration from the measured EMF potential.

## 4. Conclusions

Because of its high economic impact and demand, the efficient extraction of magnesium has recently been paid significant amounts of attention. An environmentally sound process for the extraction of magnesium has been proposed based on electrolysis reduction in the molten fluoride system. An electrochemical oxygen sensor was adopted in this study in order to propose a technique for evaluating the concentration of dissolved MgO in the molten fluoride system. In order to determine the solubility of MgO, the oxygen concentration of the $32.8MgF_2$–$67.2LiF$ system was measured using a combustion analyzer by varying the reaction time at 1083 and 1123 K. MgO saturation in the $32.8MgF_2$–$67.2LiF$ system occurred less than 2 h after the beginning of the reaction. It was also found that increasing the temperature results in a higher solubility of MgO in the fluoride system. The MgO solubility was then compared with results using other fluoride systems reported previously. It was found that the MgO solubility was not significantly affected by substituting $MgF_2$ with $CaF_2$ or changing the concentration of LiF but was mainly affected by temperature. The applicability of the EMF potential for evaluating MgO solubility was also discussed from a thermodynamic perspective. The use of the EMF cell for the quantitative measurement of MgO solubility in the molten fluoride system was evaluated. A straight calibration line was obtained, describing the relationship between the measured EMF and the logarithm of MgO concentration. Based on the obtained calibration line, the concentration of 0.4 wt% MgO system was evaluated. The calculated MgO concentration from the measured EMF potential was 0.44 wt% which was in excellent accordance with the controlled MgO concentration of 0.4 wt%. By adopting highly durable MSZ or YSZ in the solid electrolyte cell, the in-situ monitoring and evaluation of MgO solubility would be practically feasible during the electrolysis magnesium reduction process.

**Author Contributions:** Conceptualization, Y.K. and J.K.; methodology, Y.K.; formal analysis, Y.K. and J.Y.; investigation, Y.K.; data curation, Y.K.; writing–original draft preparation, Y.K.; writing–review and editing, Y.K.; supervision, J.K.; project administration, J.K. All authors have read and agreed to the published version of the manuscript.

**Funding:** The authors acknowledge financial supports from the Basic Research Project No. GP2020-013 of the Korea Institute of Geoscience and Mineral Resources (KIGAM), funded by the Ministry of Science and ICT of Korea. This work was also supported by Project No. 20000970 (NP2018-030) funded by Ministry of Trade, Industry and Energy, Republic of Korea.

**Conflicts of Interest:** The authors declare no conflict of interest.

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
