# Peer review of "Applicability of the Electrochemical Oxygen Sensor for In-Situ Evaluation of MgO Solubility in the MgF2–LiF Molten Salt Electrolysis System"

_metals, doi:10.3390/met10070906_

Round 1

Author Response

Dear Reviewer.

The authors deeply appreciate the valuable and constructive comments for improving the quality of the manuscript. Thank you for taking the time and effort in reviewing our paper

Please, kindly find the attachment file.

Reviewer 2 Report

Overall, your publication is well written. I would, however, like to see some data on reproducibility as it looks as there are only one experiment per datapoint, i.e., no uncertainty in the measurements are reported. It is not stated if you use wt% or mol% in your system, this should not be left to the reader to find out. It may not be a problem, but it should be checked if some alumina dissolves from the cell parts as this may affect the oxide solubility measurements (may easily be checked by a chemical analysis to see if Al is present).

I recommend that you use a space between a number and the % sign as this is in accordance with the international system of units (SI) and NIST.

A few minor corrections:

In line 72, the period should be placed after the reference numbers.

In line 268, the comma after Fig. 6 should be removed.

In line 273, concentration is misspelt.

Author Response

(The authors gave the same response as above.)

Reviewer 3 Report

Thanks for submitting to Metals. A new method for analyzing the MgO content in a molten salt system has been presented in this study. Detail experimental work and precise results have been shown. Though the paper is acceptable for publication, it would make a better understanding if the followings could be further addressed. Hope my comments would be of help.

1. The authors determined this method using oxygen sensor as an in-situ analysis, however, whether the reference electrode of the sensor could be affected by the given voltage in a salt pool during on-going electrochemical process is not clear. If the analysis using oxygen sensor is a just tool to find a optimum salt composition which has a large MgO solubility, I could understand. But the word 'in-situ' confuses me under what condition is this method to be applied. 

2. As a continue to Fig.6, it would more convincible if the accuracy of this new method could be shown. For such purpose, I suggest a comparison between the MgO content analyzed by the new method and the existing method as you have used to determine the equilibria time. 

A detail point:

3. P2, L73, is it the reference No. 9 correct here?

Author Response

(The authors gave the same response as above.)

Round 2

Reviewer 2 Report

Thank you for considering may recommendations for improvements.

Reviewer 3 Report

Thanks for the quick response. I think the manuscript has been well revised and no further revision would be necessary.